# A Novel Pathway of Chlorimuron-Ethyl Biodegradation by *Chenggangzhangella methanolivorans* Strain CHL1 and Its Molecular Mechanisms

**DOI:** 10.3390/ijms23179890

**Published:** 2022-08-31

**Authors:** Zhixiong Yu, Yumeng Dai, Tingting Li, Wu Gu, Yi Yang, Xiang Li, Pai Peng, Lijie Yang, Xinyu Li, Jian Wang, Zhencheng Su, Xu Li, Mingkai Xu, Huiwen Zhang

**Affiliations:** 1Key Laboratory of Pollution Ecology and Environmental Engineering, Institute of Applied Ecology, Chinese Academy of Sciences, Shenyang 110016, China; 2Basic Medical College, Shenyang Medical College, Shenyang 100034, China; 3University of Chinese Academy of Sciences, Beijing 100049, China; 4Shenyang Research Institute of Chemical Industry, Shenyang 110021, China

**Keywords:** microorganism degradation, sulfonylurea herbicide, contaminated environment, transcriptome, gene editing

## Abstract

Chlorimuron-ethyl is a widely used herbicide in agriculture. However, uncontrolled chlorimuron-ethyl application causes serious environmental problems. Chlorimuron-ethyl can be effectively degraded by microbes, but the underlying molecular mechanisms are not fully understood. In this study, we identified the possible pathways and key genes involved in chlorimuron-ethyl degradation by the *Chenggangzhangella methanolivorans* strain CHL1, a Methylocystaceae strain with the ability to degrade sulfonylurea herbicides. Using a metabolomics method, eight intermediate degradation products were identified, and three pathways, including a novel pyrimidine-ring-opening pathway, were found to be involved in chlorimuron-ethyl degradation by strain CHL1. Transcriptome sequencing indicated that three genes (*atzF*, *atzD*, and *cysJ*) are involved in chlorimuron-ethyl degradation by strain CHL1. The gene knock-out and complementation techniques allowed for the functions of the three genes to be identified, and the enzymes involved in the different steps of chlorimuron-ethyl degradation pathways were preliminary predicted. The results reveal a previously unreported pathway and the key genes of chlorimuron-ethyl degradation by strain CHL1, which have implications for attempts to enrich the biodegradation mechanism of sulfonylurea herbicides and to construct engineered bacteria in order to remove sulfonylurea herbicide residues from environmental media.

## 1. Introduction

Chlorimuron-ethyl, a sulfonylurea herbicide, is used extensively to inhibit the growth of broadleaved weeds in soybean crops [1]. However, the long-term application of chlorimuron-ethyl has led to residues in farmland soil and irrigation water, which disturb the growth of sensitive succeeding crops and other non-target plants [2,3]. Chlorimuron-ethyl inhibits acetolactate synthase, which is present in plants and microorganisms, so chlorimuron-ethyl residues in environmental media are toxic to microorganisms in soil and affect the microbial community structure [4,5,6]. It is therefore necessary to decrease chlorimuron-ethyl concentrations in environmental media.

The degradation and removal of herbicides from the environment through the use of indigenous microbial strains is an effective method [7]. Indigenous microorganisms that remove herbicides from soil and water environments have been isolated by enrichment culture. Some microorganisms can degrade chlorimuron-ethyl, and these functional microorganisms can be used to effectively bioremediate soil contaminated with chlorimuron-ethyl [8,9,10]. Several microorganisms that can degrade chlorimuron-ethyl have been enriched and isolated, and some conceivable pathways for chlorimuron-ethyl degradation have been identified [11,12]. For example, *Rhodococcus erythropolis* D310-1 has been found to degrade chlorimuron-ethyl through three pathways, the cleavage of the sulfonylurea bridge, the ester bond, and the methoxy group [13]. *Aspergillus flavus*, *Aspergillus niger*, and *Saccharomyces cerevisiae* F8 can degrade chlorimuron-ethyl by cleaving the sulfonylurea bridge and the ester bond [14]. Functional enzymes involved in chlorimuron-ethyl biodegradation have recently been identified. For example, carboxyesterase in *R. erythropolis* strain D310-1 can transform chlorimuron-ethyl into the acid precursor form [15]. Glutathione-S-transferase (GST) in *Klebsiella jilinsis* strain 2N3 can cleave the sulfonylurea bridge in chlorimuron-ethyl [16]. It has been predicted that two hydrolases (ORF0492 and ORF0934), cytochrome P450 (Kj-CysJ), and alkanesulfonate monooxygenase (Kj-SsuD) in *K. jilinsis* strain 2N3 participate in chlorimuron-ethyl biodegradation [17,18]. Despite these results, the molecular mechanisms involved in chlorimuron-ethyl biodegradation are still not fully understood, and further investigations are required. At present, the system biology-based approaches for herbicide degradation pathways and key genes mainly include genomics, transcriptomics, and metabolomics. It is efficient to determine key degradation genes through the whole-genome sequencing and transcriptome sequencing of degrading strains [17,18,19]. Metabolomics studies are mainly conducted using high-performance liquid chromatography (HPLC), liquid chromatography–mass spectrometry (LC/MS), liquid chromatography–tandem mass spectrometry (LC-MS/MS), infrared (IR) spectra, and nuclear magnetic resonance (NMR) spectra methods to predict intermediate degradation products and their degradation pathways [20].

In a previous study, we isolated and characterized a novel microorganism that degrades sulfonylurea herbicides. This was the *Chenggangzhangella methanolivorans* strain CHL1, which is a new species in a new genus in the Methylocystaceae family [21]. Strain CHL1 can efficiently degrade chlorimuron-ethyl, tribenuron-methyl, and metsulfuron-methyl [9]. Adding strain CHL1 to soil contaminated with sulfonylurea herbicides can alleviate disruption to the microbial community and decrease the toxic effects of sulfonylurea herbicides on microorganisms in the soil [6,9,22,23]. To better verify the remediation ability of strain CHL1 and evaluate its application potential in the environment, it is necessary to further study the mechanism of its degradation of chlorimuron-ethyl.

In this study, HPLC, LC/MS, and LC-MS/MS were used to preliminarily identify and quantify the products of the degradation of chlorimuron-ethyl by strain CHL1 to allow for possible degradation pathways to be identified. The gene expression changes during the degradation of chlorimuron-ethyl by strain CHL1 were characterized by transcriptome sequencing (RNA-Seq). The genes predicted to be involved in chlorimuron-ethyl degradation were confirmed using real-time quantitative reverse transcription polymerase chain reaction (qRT-PCR), gene knockout, and gene complementation techniques. The enzymes that may be involved in chlorimuron-ethyl degradation by strain CHL1 were speculated upon. The results provide new insights into the mechanisms involved in the degradation of sulfonylurea herbicides and contribute considerably to the potential environmental remediation efforts of strain CHL1.

## 2. Results and Discussion

### 2.1. Medium and Detection Conditions

Strain CHL1 grew rapidly in MSM I and MSM II media during the cultivation time (1–8 d) but grew slowly in MSM III. On day 8, chlorimuron-ethyl was not detected in MSM I or MSM II, but 28.56% of the chlorimuron-ethyl remained in MSM III. This indicates that the growth rates and chlorimuron-ethyl degradation rates of strain CHL1 were higher in MSM I and in MSM II than in MSM III (Appendix A). The yeast extract added to MSM II caused the intermediate products to be more complicated than the intermediate products in MSM I, making it more difficult to detect all of the metabolites in MSM II than in MSM I. Only some of the intermediate products in MSM III were detected, as shown in Appendix A. MSM I was therefore used in tests performed to allow for the degradation metabolites of chlorimuron-ethyl to be identified.

The degradation metabolites of chlorimuron-ethyl were detected using two HPLC methods (LCI and LCII). Three compounds with symmetry factors of 0.96 were detected using LC I, and six compounds with symmetry factors of 0.99 were detected using LC II. The peak shapes and sensitivities were markedly better using LC II than LC I, as shown in Appendix A.

### 2.2. Pathways and Mechanisms Involved in Chlorimuron-Ethyl Degradation by Strain CHL1

Eight intermediate degradation products were detected during the degradation of chlorimuron-ethyl by strain CHL1. The mass-to-charge ratio for the chlorimuron-ethyl [M + H]^+^ ion was 414.93, and the mass-to-charge ratios for the eight intermediate degradation product [M + H]^+^ ions were 274.16 (product I), 159.97 (product II), 386.19 (product III), 202.06 (product IV), 185.81 (product V), 340.15 (product VI), 302.07 (product VII), and 284.19 (product VIII), as shown in Figure 1 and Appendix A.

The intermediate degradation products that were found led us to propose three pathways for the degradation of chlorimuron-ethyl by strain CHL1. The pathways are shown in Figure 2. Pathway I is initiated by the cleavage of the sulfonylurea bridge in chlorimuron-ethyl to give products I and II. Product I is then de-esterized and decarboxylated to give product IV. Pathway II involves the de-esterification of the benzene ring side chain of chlorimuron-ethyl to give product III. Products II and IV are given by the cleavage of the sulfonylurea bridge in product III. The alternative pathway III gives product VI through the opening of the pyrimidine ring and the demethylation and dechlorination of product III. Product VI is then degraded stepwise to give product IV, through dealkylation to give product VII, dehydroxylation to give product VIII, and the cleavage of the sulfonylurea bridge to give product IV. Pathways I, II, and III each contain a degradation step in which product IV is transformed into the final product V through a cyclization–dehydration reaction.

The cleavage of the sulfonylurea bridge, de-esterification, and cyclization–dehydration are common pathways for chlorimuron-ethyl degradation by microbes [13,14]. Pyrimidine ring opening has not previously been reported as being a pathway for chlorimuron-ethyl degradation by strain CHL1. Pyrimidine ring opening gives the intermediate degradation products VI and VII and product VIII. The end product of chlorimuron-ethyl degradation by strain CHL1 is 3-hydroxy-2,3-dihydro-1λ6,2-benzothiazole-1,1-dione (product V), but the end product of chlorimuron-ethyl degradation by other strains is o-sulfonate benzoic imide [13,14]. Chlorimuron-ethyl degradation by different strains can give different end products. In the novel pathway, the pyrimidine ring in chlorimuron-ethyl is completely degraded.

### 2.3. RNA-Seq Analysis of Strain CHL1 during Chlorimuron-Ethyl Degradation

This is example 1 of an equation: Totals of 14,940,667 bp, 22,600,503 bp, and 22,664,187 bp clean reads were obtained for samples A, B, and C, respectively, in the control group. Totals of 14,118,074 bp, 21,687,283 bp, and 21,844,017 bp clean reads were obtained for samples A, B, and C, respectively, in the treatment group. More transcriptome details are given in Appendix A.

The mechanisms involved in chlorimuron-ethyl degradation were investigated by performing a global transcriptome analysis of strain CHL1 grown in the absence and presence of chlorimuron-ethyl in the early (sample A), middle (sample B), and late (sample C) exponential growth phases. In samples A, B, and C, 34 genes (eight up-regulated and 26 down-regulated), 1446 genes (702 up-regulated and 744 down-regulated), and 2226 genes (1037 up-regulated and 1189 down-regulated), respectively, were found to be differentially expressed genes (DEGs) in the control and treatment groups, as shown in Figure 3a–c. A total of 2725 genes were found to be DEGs during the degradation of chlorimuron-ethyl by strain CHL1.

The main categories of up-regulated DEGs were for the serine family amino acid catabolic process, glycine catabolic process, glycine decarboxylation via glycine cleavage system, and divalent inorganic cation transmembrane transporter activity, which could be associated with chlorimuron-ethyl degradation, as shown in Figure 3d. The down-regulated DEGs were mainly enriched in the category of small protein activating enzyme activity and were probably affected by the toxicities of chlorimuron-ethyl and the products, as shown in Figure 3e.

In a previous study, it was found that “xenobiotic biodegradation and metabolism” and “sulfur metabolism” significantly correlated with the degradation of chlorimuron-ethyl [18]. In this study, the expressions of *atzF* and *atzD* (in the atrazine degradation of xenobiotic biodegradation and metabolism category) and *cysJ* (in the sulfur metabolism category) in strain CHL1 were significantly up-regulated (*p* < 0.05), as shown in Table 1. We therefore hypothesized that *atzF*, *atzD*, and *cysJ* may be involved in chlorimuron-ethyl degradation by strain CHL1.

### 2.4. Molecular Engineering

We used a Lambda-Red system to genetically manipulate strain CHL1. Because of the results of the previous analyses, we selected three genes (*atzF*, *atzD*, and *cysJ*) for use in gene knock-out and complementation experiments. The chlorimuron-ethyl degradation rates were 17.7%, 13.1%, and 48.4% lower (all significantly lower; *p* < 0.05) for the knockout strains CHL1∆*atzF*, CHL1∆*atzD*, and CHL1∆c*ysJ*, respectively, than the wild strain CHL1 on day 7, as shown in Figure 4. The chlorimuron-ethyl degradation rate remained >50% after knocking out the *atzF*, *atzD*, or *cysJ* gene in strain CHL1. This could have been caused by the presence of isoenzymes or there being multiple pathways for chlorimuron-ethyl degradation by strain CHL1 [14]. To avoid biomass interfering with the knockout strains, the chlorimuron-ethyl degradation rates were normalized to the biomass concentration expressed as the optical density at 600 nm. The normalized chlorimuron-ethyl degradation rates were lower for all of the knockout strains than the wild strain (*p* < 0.05), consistent with the results of the tests described above (Appendix A). This indicates that the decreased abilities of the three mutant strains to degrade chlorimuron-ethyl were caused by knocking out the genes rather than decreased growth. The chlorimuron-ethyl degradation rates for the three complemented strains CHL1Δ*atzF*[pEG-*atzF*], CHL1Δ*atzD*[pEG-*atzD*], and CHL1Δ*cysJ*[pEG-*cysJ*] were similar to the chlorimuron-ethyl degradation rates for the wild strain CHL1 (*p* > 0.05), as shown in Figure 4. These results indicate that *atzF*, *atzD*, and *cysJ* directly or indirectly participate in chlorimuron-ethyl degradation by strain CHL1.

### 2.5. Real-Time Quantitative Reverse Transcription PCR (qRT-PCR)

The transcriptome data were verified by qRT-PCR, and the mRNA expression levels for the genes involved in chlorimuron-ethyl degradation were determined at culture times between 1 and 7 d. The results are shown in Figure 5. The results indicate that the mRNA expression levels of *atzF*, *atzD*, and *cysJ* were significantly higher for the treatment group than for the control group on days 6 and 7 (*p* < 0.05). The qRT-PCR results for the three genes involved in chlorimuron-ethyl degradation were consistent with the RNA-Seq results. These results suggest that *cysJ*, *atzD*, and *atzF* participate in different steps of the mechanism involved in chlorimuron-ethyl degradation by strain CHL1.

### 2.6. Structural Analysis

The nucleotide sequences for *cysJ*, *atzD*, and *atzF* were submitted to the NCBI GenBank, and the accession numbers are OK050583, OK050585, and OK050586, respectively. Genes *atzD*/*atzF* and *cysJ* are in different gene clusters, as shown in Figure 6a. CysJ has a molecular weight of 44.63 kDa and a pI of 6.44. AtzD has a molecular weight of 26.08 kDa and a pI of 9.92. AtzF has a molecular weight of 61.12 kDa and a pI of 5.05. CysJ has a secondary structure containing 35.87% α-helices, 4.75% β-turns, and 52.49% random coils. AtzF has a secondary structure containing 37.98% α-helices, 3.98% β-turns, and 43.95% random coils. AtzD has a secondary structure containing 33.47% α-helices, 8.76% β-turns, and 41.83% random coils.

The tertiary structures of the enzymes CysJ, AtzD, and AtzF predicted using SWISS-MODEL are shown in Appendix A. The reference tertiary structure models of AtzF, AtzD, and CysJ were 6cwj.1, 3a1i.1, and 6j7i.1, respectively. From the tertiary structures of the proteins, the predicted binding sites for chlorimuron-ethyl were Asp82-Lys199-Gly200-Val201 in AtzF (Figure 6b), Glu43-Asn49-Thr80 in AtzD (Figure 6c), and Trp254-Gly322-Asp325 in CysJ (Figure 6d). In future experiments, the key active amino acid sites for chlorimuron-ethyl binding will be identified by site-directed mutation. The key active amino acid sites will be engineered to enhance the activities of the enzymes for the degradation of chlorimuron-ethyl [24].

### 2.7. Predicted Mechanisms for Enzymatic Degradation of Chlorimuron-Ethyl

AtzF had a high degree of homology with allophanate hydrolase produced by *Hansschlegelia quercus*, and the amino acid sequences had the highest similarity value (67.99%), as shown in Appendix A. Allophanate hydrolase AtzF can degrade atrazine by attacking the carbon–nitrogen bond, so we speculated that AtzF attacks the carbon–nitrogen bond in the sulfonylurea bridge of chlorimuron-ethyl [25]. AtzD had a high degree of homology with cyanuric acid amidohydrolase produced by *Rhizobium* sp. BK376, as shown in Appendix A. Cyanuric acid amidohydrolase can open the triazine ring in atrazine, and we speculated that AtzD could participate in the opening of the pyrimidine ring in chlorimuron-ethyl [26]. CysJ had a high degree of homology with the NAD(P)H-dependent nitrite reductase flavoprotein subunit produced by *Methylopila* sp. strain Yamaguchi, as shown in Appendix A. CysJ was found to contain cytochrome p450-like alpha subunits in the domain. Enzyme Kj-CysJ produced by *K. jilinsis* 2N3 can degrade chlorimuron-ethyl and contains cytochrome p450-like alpha subunits in the domain, but the mechanism involved is not clear. Cytochrome P450 can degrade chlorimuron-ethyl by catalyzing the carbon–oxygen bond, so CysJ was predicted to demethylate chlorimuron-ethyl [27].

### 2.8. Schemes of the Functional Genes Involved in the Pathways Involved in Chlorimuron-Ethyl Degradation by Strain CHL1

We predicted the enzymes through which chlorimuron-ethyl is degraded by strain CHL1, as shown in Figure 7. Strain CHL1 contains a complete ABC transporter system, which has been found to be important for the elimination of toxic compounds (xenobiotics) [28]. Chlorimuron-ethyl was therefore predicted to be imported into strain CHL1 cells by ABC transporters.

We predicted that AtzF cleaves the sulfonylurea bridge in chlorimuron-ethyl, that AtzD participates in opening the pyrimidine ring in chlorimuron-ethyl, and that CysJ demethylates chlorimuron-ethyl. It was previously found that esterase SulE in strain CHL1 can cleave the ester bond in chlorimuron-ethyl [29,30,31]. The transcriptome results indicated that the genes encoding glutathione S-transferase (GST), dehalogenase (CopA), desaturase (CrtD), hydratase (CrtC), and monooxygenase (Limb) were markedly up-regulated during the degradation of chlorimuron-ethyl by strain CHL1, as shown in Appendix A. It has previously been found that GST can cleave the sulfonylurea bridge [16], CopA can transport ions [32], CrtD can affect CH–CH group donors [33], CrtC can cleave carbon–oxygen bonds [34], and Limb can affect paired donors to incorporate or reduce molecular oxygen [35]. The metabolic pathway and expression of genes for the degradation of chlorimuron-ethyl by strain CHL1 indicated that five enzymes (GST, CopA, CrtD, CrtC, and Limb) may be involved in chlorimuron-ethyl degradation.

In summary, nine enzymes (AtzF, AtzD, CysJ, SulE, GST, CopA, CrtD, CrtC, and Limb) were predicted to be involved in the degradation of chlorimuron-ethyl by strain CHL1. In pathway I, product IV is produced from chlorimuron-ethyl by AtzF and SulE. In pathway II, product IV is produced from chlorimuron-ethyl by SulE and GST. In pathway III, product IV is produced from chlorimuron-ethyl by SulE, CysJ, AtzD, CopA, CrtD, CrtC, and GST. The transformation of product IV into the final product V by Limb occurs in all three pathways. The results indicate that AtzF, AtzD, CysJ, and SulE are involved in the degradation of chlorimuron-ethyl. However, the mechanisms involved in chlorimuron-ethyl degradation by AtzF, AtzD, CysJ, and SulE need to be further explored. The abilities of other enzymes (GST, CopA, CrtD, CrtC, and Limb) to degrade chlorimuron-ethyl remain to be studied.

## 3. Materials and Methods

### 3.1. Plasmids and Bacterial Strains

The plasmids and bacterial strains used in this study are listed in Appendix A. The primer sequences designed in this study are shown in Appendix A. *Chenggangzhangella methanolivorans* strain CHL1, was screened, isolated, and preserved in our laboratory, and the preservation number is CGMCC11649 [10,21].

### 3.2. Medium and Detection Conditions

Chlorimuron-ethyl was added to mineral salts medium (MSM) I, II, and III (Appendix A) at a final concentration of 1000 mg·L^−1^ without and with strain CHL1 in order to culture for 8 days (30 °C, 180 rpm). OD_600_ values were measured at 24 h intervals. The residual chlorimuron-ethyl in the collected samples on the eighth day was quantified using high-performance liquid chromatography (HPLC) equipped with a Zorbax C-18 ODS Spherex column (4.6 × 250 mm, 5 μm, Agilent Technologies, Palo Alto, CA, USA), and the conditions of the liquid chromatography (LC) were LC I and LC II. The conditions of liquid chromatography (LC) I were as follows: 0.5% acetic acid: methanol (30:70, *v*/*v*), column temperature 25 °C [8]. The conditions of LC II were as follows: acetonitrile: 0.2% acetic acid, column temperature 22 °C 0–1 min: 2% acetonitrile, 1–10 min: 2–70% acetonitrile, 10–13 min: 70–100% acetonitrile, 13–13.5 min: 100–2% acetonitrile.

### 3.3. Pathways and Mechanisms of Chlorimuron-Ethyl Degradation

Chlorimuron-ethyl was added to MSM I at a final concentration of 2000 mg·L^−1^ without and with strain CHL1 to culture for 30 days (30 °C, 180 rpm). All culture supernatant samples were collected in triplicate every day. After being filtrated with 0.22 μm nylon filters, the collected samples were analyzed using HPLC equipped with a Zorbax C-18 ODS Spherex column (4.6 × 250 mm, 5 μm), liquid chromatography–mass spectrometry (LC/MS) equipped with a Waters Atlantis T3 (2.1 mm × 150 mm), and liquid chromatography–tandem mass spectrometry (LC-MS/MS) equipped with an electrospray ionization (ESI) (Appendix A). The conditions of the high-performance liquid chromatography (HPLC) were as follows: 20 μL of solution was injected into an HPLC equipped with a Zorbax C-18 ODS Spherex column (4.6 × 250 mm, 5 μm, Agilent Technologies, Palo Alto, CA, USA) and separated at a flow rate of 1 mL·min−1. Chlorimuron-ethyl was detected at 254 nm [10]. The conditions of the liquid chromatography–mass spectrometry (LC/MS) were as follows: 10 μL solution was injected into a Thermo Finnigan LCQ Deca LC/MS n system (Thermo LCQ DECA Xp MAX, Thermo Finnigan, MA, USA) equipped with a Waters Atlantis T3 (2.1 mm × 150 mm) and a thermostat (20 °C) at flow rate of 0.25 mL·min^−1^. Methanol (phase A), acetonitrile (phase B), and 5 mmol·L-1 ammonium acetate (phase C) were the mobile phases. The procedure was: 0–14 min: 10%A + 10%B + 80%C; 14–16 min: 45%A + 45%B + 10%C; 16–18 min: 48%A + 48%B + 4%; 18–20 min: 10%A + 10%B + 80%C. M/Z 50 to 500 [8]. The conditions of the liquid chromatography–tandem mass spectroscopy (LC-MS/MS) were as follows: the MS apparatus was equipped with an electrospray ionization (ESI): spray voltage, 4.5 kV; capillary temperature, 350 °C; drying gas (nitrogen) flow rate, 10 L min^−1^; nebulizer gas pressure, 35.0 psi; and mass range, M/Z 50–500 [36].

### 3.4. Transcriptome Sequencing

Based on the growth and chlorimuron-ethyl degradation of strain CHL1, we sampled the early (sample A), middle (sample B), and late (sample C) periods of the fastest degradation period of chlorimuron-ethyl to test the transcriptome [37,38]. Strain CHL1 was cultured in MSM I with chlorimuron-ethyl-free and chlorimuron-ethyl (2000 mg·L^−1^) for 7 days (30 °C, 180 rpm). The bacterial precipitations were collected by centrifugation (8000 rpm, 10 min) and frozen in liquid nitrogen and stored at −80 °C. All assays had three replicates. The samples were treated by Personal Biotechnology Co., (Shanghai, China) for transcriptome sequencing (Appendix A).

### 3.5. Differentially Expressed Genes (DEG) Analysis

Gene expression levels were analyzed with RNA-Seq by Expectation Maximization (RSEM) and quantified using Transcripts Per Million reads (TPM) value, and the complete genome sequence of strain CHL1 was performed as a reference [39]. To obtain a global view of the transcriptome in strain CHL1 for chlorimuron-ethyl degradation, we analyzed the DEGs using the DESeq2 package [40]. The gene expression levels identified in the RNA-Seq sequence were quantified by the value of Log_2_FC into three groups. Genes with expression levels of Log_2_FC = 0, Log_2_FC < 0, and Log_2_FC > 0 showed no change, down-regulated change, and up-regulated change, respectively. The value of the false discovery rate (FDR) of ≤0.05 and |Log_2_FC |> 0 was used to screen significant DEGs.

### 3.6. Molecular Engineering

Three predicted functional genes (*atzF*, *atzD*, *cysJ*) were knocked out in strain CHL1 using the Lambda-Red system (Appendix A). The plasmid pEarleyGate100 was used to construct the corresponding complementary strains of three chlorimuron-ethyl degradation genes (Appendix A). All the mutant strains were 1% inoculated into 100 mL of MSM I with 100 mg·L^−1^ chlorimuron-ethyl to culture for 7 days (30 °C, 180 rpm). The bacterial growth (OD_600_) and concentration of residual chlorimuron-ethyl were detected every day.

### 3.7. Real-Time Quantitative Reverse Transcription PCR (qRT-PCR) Assay

We performed qRT-PCR with PowerSYBR^®^Green PCR Master Mix to quantify three chlorimuron-ethyl degradation genes (*atzF*, *atzD*, *cysJ*), and the 16S rRNA gene was used as a reference. Gene expression relative to that of the reference was analyzed using the 2^−ΔΔCT^ method [36]. Total RNA was extracted and purified using a Bacteria Total RNA Isolation Kit (manufacturer), and cDNA was synthesized from total RNA using a PrimeScript TM RT reagent Kit with gDNA Eraser [18]. All assays were performed in three replicates.

### 3.8. Structure and Function Prediction

The nucleotide and amino acid sequence analyses of AtzF, AtzD, and CysJ were performed using DNAMAN [41]. The conserved domain analyses were performed with the Conserved Domain Database (CDD) in the National Center for Biotechnology Information (NCBI), and the tertiary structure was predicted using SWISS-MODEL [42,43]. Chlorimuron-ethyl substrate was docked into a protein model by AutoDock Vina with default settings [44]. The productive pose was determined based on the docking score. The structure models were visualized using the PyMOL program [45]. The amino acid sequences of AtzF, AtzD, and CysJ were blasted by Universal Protein (Uniprot), and the top 10 sequences were selected for a molecular phylogenetic analysis using MEGA 7.0 software [41].

### 3.9. Statistical Analysis

Independent-sample *t*-tests were performed to determine significant differences using SPSS 23.0. *p*-value < 0.05 was considered a significant difference.

## 4. Conclusions

We speculated that the degradation of chlorimuron-ethyl by strain CHL1 involves three metabolic pathways and eight metabolites. Different levels of gene expression by strain CHL1 in different growth phases (early, middle, and late logarithmic growth phases) in the presence of chlorimuron-ethyl were found using the RNA-Seq technique. A total of 2725 genes were found to be DEGs during the degradation of chlorimuron-ethyl by strain CHL1. Gene knockout, gene complementation, and qRT-PCR investigations indicated that *atzF*, *atzD*, and *cysJ* are involved in the degradation of chlorimuron-ethyl by strain CHL1. The trends in mRNA expression determined by qRT-PCR are consistent with the RNA-Seq results. Compared with the wild strain CHL1, the degradation ability of the knockout strains CHL1∆*atzF*, CHL1∆*atzD*, and CHL1∆*cysJ* significantly reduced (*p* < 0.05), and their complement strains hardly changed (*p* > 0.05). The pathways involved in chlorimuron-ethyl degradation by strain CHL1 and the gene expression levels indicated that the enzymes AtzF, AtzD, CysJ, SulE, GST, CopA, CrtD, CrtC, and Limb may be the key to chlorimuron-ethyl degradation by strain CHL1. The functions of the enzymes AtzF, AtzD, CysJ, and SulE were confirmed, and the abilities of the enzymes GST, CopA, CrtD, CrtC, and Limb to degrade chlorimuron-ethyl are yet to be studied. The results improve our understanding of the mechanisms involved in chlorimuron-ethyl degradation and provide reference data for future investigations of the biodegradation of sulfonylurea herbicides. Many other genes that may be involved in the degradation of chlorimuron-ethyl by strain CHL1 are currently being investigated to better elucidate the biodegradation mechanism.

## Figures and Tables

**Figure 1 ijms-23-09890-f001:**
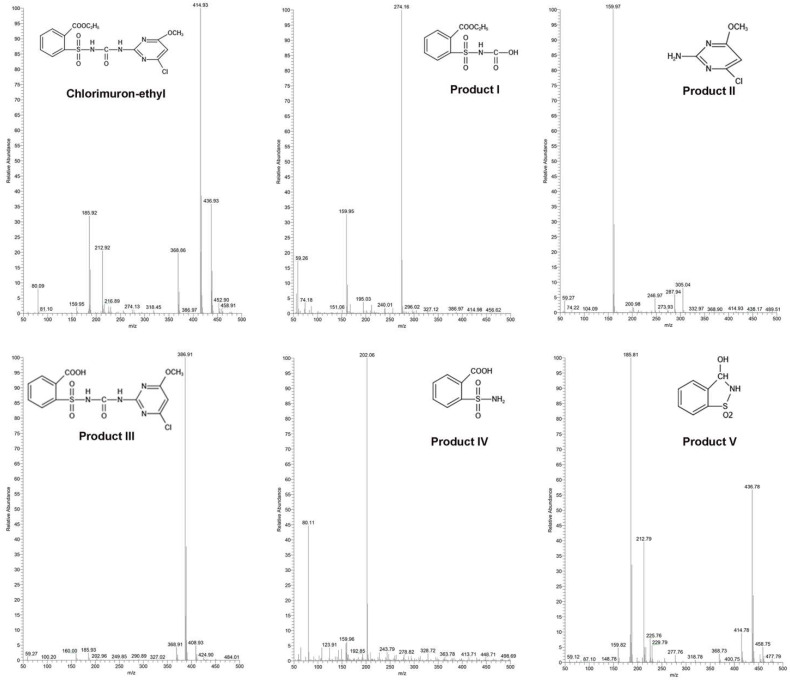
Mass spectra of chlorimuron-ethyl and chlorimuron-ethyl degradation products by strain CHL1. All assays were performed in three replicates.

**Figure 2 ijms-23-09890-f002:**
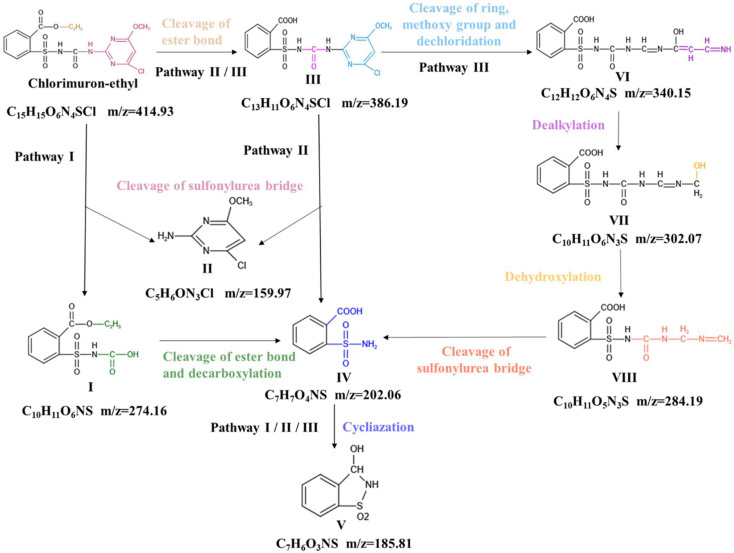
Metabolic pathways involved in chlorimuron-ethyl degradation by strain CHL1.

**Figure 3 ijms-23-09890-f003:**
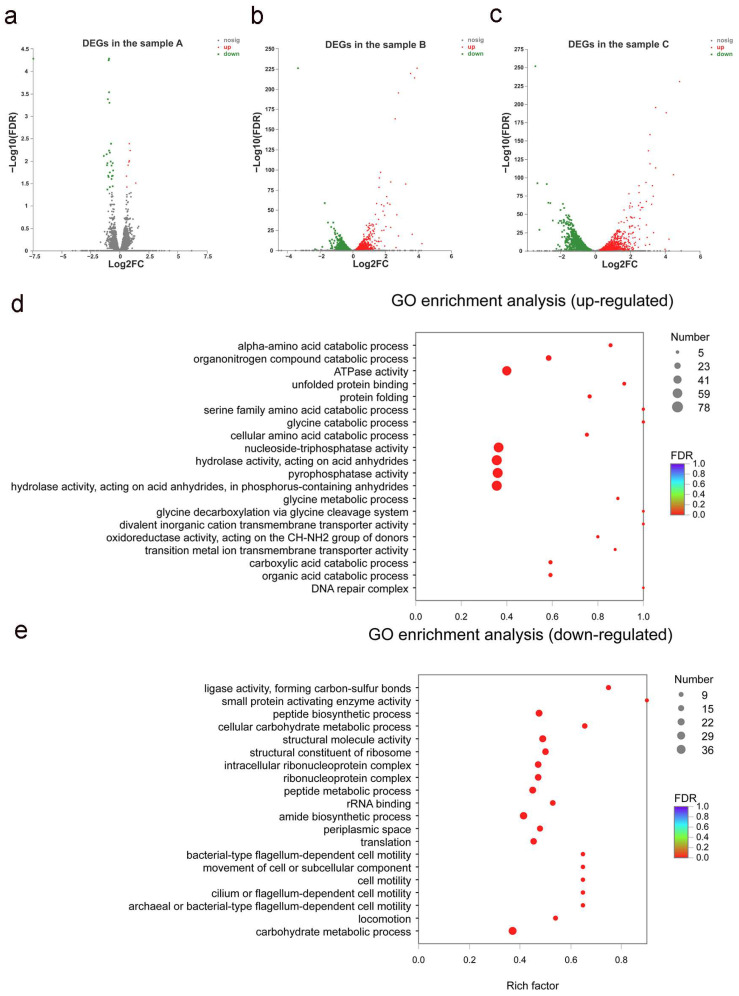
Transcriptome analysis results for strain CHL1. Differentially expressed genes in sample A (**a**), sample B (**b**), and sample C (**c**). Each symbol represents a gene. Up-regulated, down-regulated, and undifferentiated genes are shown in red, green, and black, respectively. Enrichment of up-regulated genes (**d**) and down-regulated genes (**e**) in the Gene Ontology database are shown. All assays were performed in three replicates.

**Figure 4 ijms-23-09890-f004:**
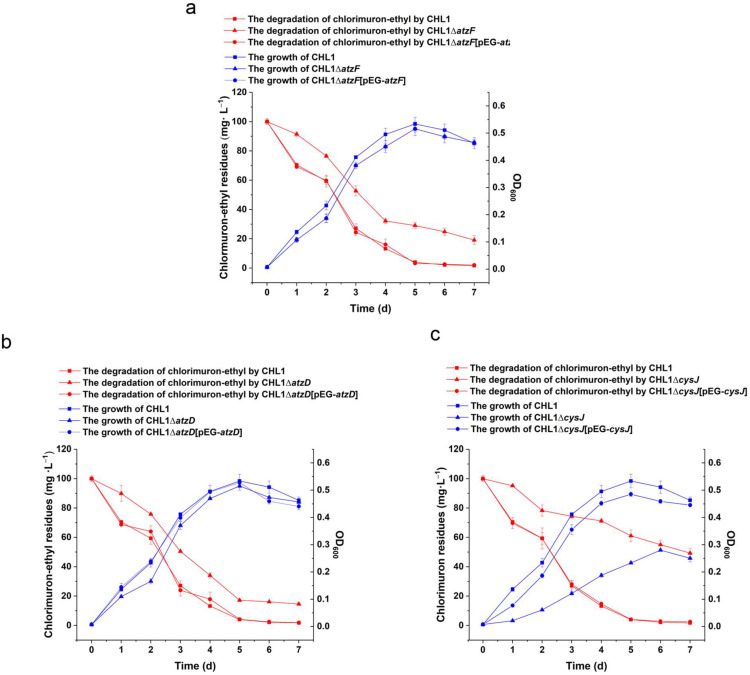
Degradation of chlorimuron-ethyl by strains with knocked out and complemented genes *atzF* (**a**), *atzD* (**b**), and *cysJ* (**c**). Blue indicates the growth curve, and red indicates degradation of chlorimuron-ethyl. All assays were performed in three replicates.

**Figure 5 ijms-23-09890-f005:**
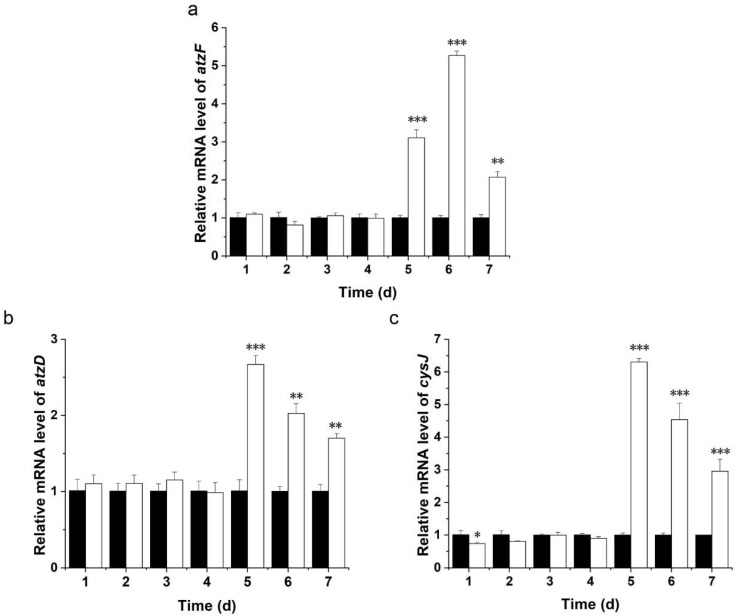
Relative mRNA expression levels of *atzF* (**a**), *atzD* (**b**), and *cysJ* (**c**) in strain CHL1 grown in the presence or absence of chlorimuron-ethyl for 1–7 d. The white and black bars represent expressions in the treatment groups and control groups, respectively. The 16S rRNA gene was used as an endogenous control. Statistically significant differences between mutant and wild strains are marked * (*p* < 0.05), ** (*p* < 0.01), or *** (*p* < 0.001). All assays were performed in three replicates.

**Figure 6 ijms-23-09890-f006:**
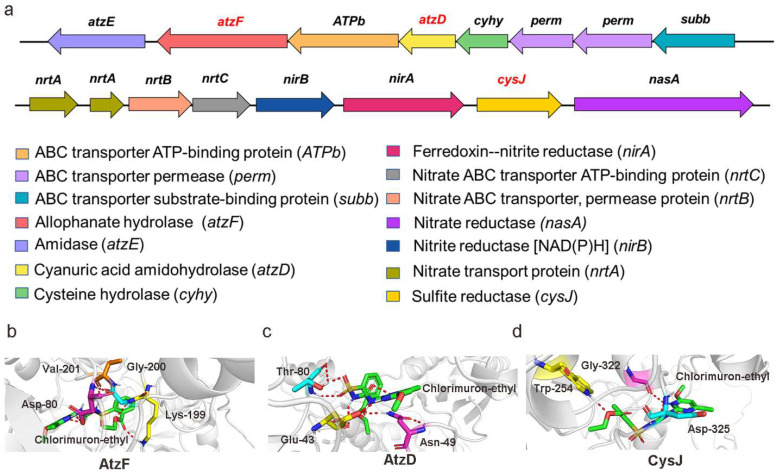
Gene clusters (**a**) and binding sites between chlorimuron-ethyl and AtzF (**b**), AtzD (**c**), and CysJ (**d**).

**Figure 7 ijms-23-09890-f007:**
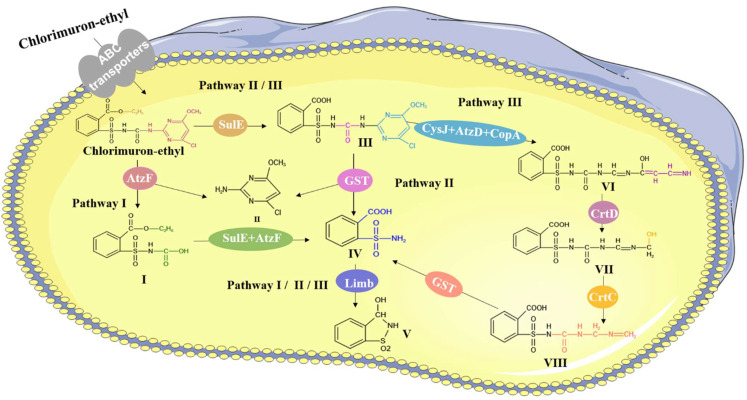
Chlorimuron-ethyl degradation pathways associated with functional genes in strain CHL1.

**Table 1 ijms-23-09890-t001:** Basic information of chlorimuron-ethyl degradation genes in strain CHL1 in this study.

Gene Name	Annotation	Log_2_FC			KEGG
		Sample A	Sample B	Sample C	
*atzF*	Allophanate hydrolase	0.23	0.82 *	1.77 *	Atrazine degradation
*atzD*	Cyanuric acid amidohydrolase	−0.65	1.74 *	2.03 *	Atrazine degradation
*cysJ*	Sulfite reductase	0.06	1.17 *	2.57 *	Sulfur metabolism

Statistically significant differences are marked with * (false discovery rate FDR < 0.05). If the value of Log_2_FC was greater than 0, it indicated up-regulation. If the value of Log_2_FC was less than 0, it indicated down-regulation.

## Data Availability

The transcriptome sequence of *Chenggangzhangella methanolivorans* strain CHL1 during the degradation of chlorimuron-ethyl was deposited in NCBI under the accession number SUB10274035.

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
