# Peer review of "A Novel Pathway of Chlorimuron-Ethyl Biodegradation by Chenggangzhangella methanolivorans Strain CHL1 and Its Molecular Mechanisms"

_ijms, 2022, doi:10.3390/ijms23179890_

Round 1

Reviewer 1 Report

Title: A Novel Pathway of Chlorimuron-ethyl Biodegradation by Chenggangzhangella methanolivorans Strain CHL1 and Its Molecular Mechanisms

Ms. Number: ijms-1854866

The authors studied and reported about a Novel Pathway of Chlorimuron-ethyl Biodegradation by Chenggangzhangella methanolivorans Strain CHL1 and Its Molecular Mechanisms. The manuscript was well organized and the data was properly discussed. The work was suitable for the publication. However, before going to accept, few corrections need to be incorporated.

Authors needs to revise the manuscript.

There are typographical and spelling mistakes that needs to be address by the authors. 

Authors can improve the quality of the figures.

In addition, authors can discuss the system biology based approaches for the herbicides degradations in Introductions.

https://pubs.acs.org/doi/abs/10.1021/jf1030339

The degradation pathway must be cross checked again.

Authors can cite some of the interesting findings on degradation and removal of herbicides from soil and water environment using indigenous microbial strains. 

Author Response

    Thank you for your comments concerning our manuscript entitled “A Novel Pathway of Chlorimuron-ethyl Biodegradation by Chenggangzhangella methanolivorans Strain CHL1 and Its Molecular Mechanisms” (Manuscript ID: ijms-1854866). Those comments and suggestions are all valuable and very helpful for revising and improving our paper, as well as the important guiding significance to our research. We have carefully studied these comments and have made correction.

1. There are typographical and spelling mistakes that needs to be address by the authors.

Reply: Thank you for your suggestion. We have checked and corrected typographical and spelling errors in the manuscript. Gene names and genus names need to be italicized on the line 180-197, 219-221, 242-252. The word “de-carboxylased” was changed to “decarboxylated” in line 114.

2. Authors can improve the quality of the figures.

Reply: Thank you for your suggestion, we have improved the quality of the figures. Fonts are enlarged for easier reading in Figures 2,4,5,7. The clarity of the figures has also been improved.

3. In addition, authors can discuss the system biology-based approaches for the herbicides degradations in Introductions. https://pubs.acs.org/doi/abs/10.1021/jf1030339

Reply: Thank you for your suggestion, we have discussed the system biology-based approaches for the herbicides degradations in Introductions in line 58-66. We have carefully read and cited this literature you shared and related literatures to supplement this part of the content.

Line 58-66: At present, the system biology-based approaches for the herbicides degradation pathways and key genes mainly included genomics, transcriptomics, and metabolomics. It was an efficient method to obtain key degradation genes through whole genome sequencing and transcriptome sequencing of degrading strains [17-19]. Metabolomics was mainly through high performance liquid chromatography (HPLC), liquid chromatography mass spec-trometry (LC/MS), liquid chromatography tandem mass spectrometry (LC-MS/MS), infra-red (IR) spectra and nuclear magnetic resonance (NMR) spectra methods to predict inter-mediate degradation products and their degradation pathways [20].

4. The degradation pathway must be cross checked again.

Reply: Thank you for your suggestion, we have cross checked the degradation pathways. The misspelling of words in the figures has been revised.

5. Authors can cite some of the interesting findings on degradation and removal of herbicides from soil and water environment using indigenous microbial strains.

Reply: Thank you for your suggestion. We have cited some findings on degradation and removal of herbicides from soil and water environment using indigenous microbial strains in line 39-42.

Line 39-42: Degradation and removal of herbicides from the environment through the use of indigenous microbial strains is an effective method [7]. Indigenous microorganisms that remove herbicides from soil and water environment have been isolated by enrichment culture.

Reviewer 2 Report

In general, this article is scientifically sound and well-written. Some comments and suggestions:

1. The authors need to describe in the methods section on the metabolomics approach being used to identify the eight products and three pathways.

2. The font for most of the figures are too small to be seen.

3. For the experiments, the number of repeats are not shown in the figure legend.

4. Is the data for the metabolomics and transcriptomics being deposited to the public database? If so, please indicate in the manuscript.

5. Did the authors use any standards to confirm the identified metabolites in their study?

6. Has any quantification of concentration been performed for the metabolites identified?

7. Can the authors identify the intermediate products in the engineered strains (knockout)? 

Author Response

    Thank you for your comments concerning our manuscript entitled “A Novel Pathway of Chlorimuron-ethyl Biodegradation by Chenggangzhangella methanolivorans Strain CHL1 and Its Molecular Mechanisms” (Manuscript ID: ijms-1854866). Those comments and suggestions are all valuable and very helpful for revising and improving our paper, as well as the important guiding significance to our research. We have carefully studied these comments and have made correction.

1. The authors need to describe in the methods section on the metabolomics approach being used to identify the eight products and three pathways.

Reply: Thanks to your suggestion, we have added the metabolomic approach to identify products and pathways in the Methods section in line 315-329.

Line 315-329: The condition of high-performance liquid chromatography (HPLC) was as follows: 20 μL of solution was injected into an HPLC equipped with a Zorbax C-18 ODS Spherex column (4.6 × 250 mm, 5 μm, Agilent Technologies, Palo Alto, CA, USA) and separated at a flow rate of 1 mL·min−1. Chlorimuron-ethyl was detected at 254 nm [10]. The condition of liq-uid chromatography mass spectrometry (LC/MS) was as follows: 10 μL solution was in-jected into a Thermo Finnigan LCQ Deca LC/MS n system (Thermo LCQ DECA Xp MAX, Thermo Finnigan, MA, USA) equipped with a Waters atlantis T3 (2.1 mm × 150 mm), and a thermostat (20 ℃) at flow rate of 0.25 ml·min-1. Methanol (phase A), acetonitrile (phase B) and 5mmol·L-1 ammonium acetate (phase C) were the mobile phases. The procedure was: 0-14min: 10%A+10%B+80%C; 14-16min: 45%A+45%B+10%C; 16-18min: 48%A+48%B+4%; 18-20min: 10%A+10%B+80%C. M/Z 50 to 500 [8]. The condition of liquid chromatography tandem mass spectroscopy (LC-MS/MS) was as follows: The MS apparatus were equipped with an electrospray ionization (ESI): spray voltage, 4.5 kV; capillary temperature, 350 ℃; drying gas (nitrogen) flow rate, 10 L min−1; nebulizer gas pressure, 35.0 psi; and mass range, M/Z 50 to 500 [36].

2. The font for most of the figures are too small to be seen.

Reply: Thanks to your suggestion, we have increased the font size of the figures for easier reading in Figures 2,4,5,7.

3. For the experiments, the number of repeats are not shown in the figure legend.

Reply: Thanks to your suggestion, we have shown the number of repetitions in Figure 1,3,4,5 legends in line 136, 165-166, 202 and 217, respectively. All assays were performed three replicates.

4. Is the data for the metabolomics and transcriptomics being deposited to the public database? If so, please indicate in the manuscript.

Reply: Thanks to your question. The transcriptomic data have been submitted in public databases. In the manuscript, we have shown that the data was deposited in NCBI under the accession number SUB10274035 in Data Availability Statement Section in line 410-412. We predicted intermediate degradation products and pathways by HPLC, LC/MS, LC/MS/MS as detailed in Figure 1 and Table S1, without depositing into public databases.

Line 412-414: Data Availability Statement: The transcriptome sequence of Chenggangzhangella methanolivorans strain CHL1 during the degradation of chlorimuron-ethyl was deposited in NCBI under the accession number SUB10274035.

5. Did the authors use any standards to confirm the identified metabolites in their study?

Reply: Thanks to your question. Due to the instability of intermediate metabolites, standards could not be prepared for corresponding validation. By HPLC, LC/MS, LC/MS/MS, the intermediate degradation products are predicted according to the mass-to-charge ratio and the structure of chlorimuron-ethyl, which is a relatively recognized and commonly used research method.

6. Has any quantification of concentration been performed for the metabolites identified?

Reply: Thanks for your question. The concentrations of intermediate degradation products are dynamic, being produced while being consumed, so their concentrations cannot be quantified.

7. Can the authors identify the intermediate products in the engineered strains (knockout)?

Reply: Thanks for your question. We have tried to identify intermediate degradation products of the engineered strains (knockout), but there is no obvious difference between intermediate degradation products of the engineered strain (knockout) and the wild strain CHL1. It may be because the intermediate degradation products are unstable and cannot be quantitatively determined.